# Tailoring Morphology in Hydrothermally Synthesized CdS/ZnS Nanocomposites for Extraordinary Photocatalytic H$_2$ Generation via Type-II Heterojunction

**Mianli Huang [1], Maoqing Yu [1], Ruiru Si [2], Xiaojing Zhao [1], Shuqin Chen [1,\*], Kewei Liu [1] and Xiaoyang Pan [1,\*]**

[1] College of Chemical Engineering and Materials, Quanzhou Normal University, Quanzhou 362000, China; mianlihuang@qztc.edu.cn (M.H.); 17705053910@163.com (M.Y.); zhaoxj199011@163.com (X.Z.); 10005@qztc.edu.cn (K.L.)

[2] Fujian Key Laboratory of Agro-Products Quality and Safety, Institute of Quality Standards & Testing Technology for Agro-Products, Fujian Academy of Agricultural Sciences, Fuzhou 350003, China; srr0506@163.com

\* Correspondence: chenshuqin@qztc.edu.cn (S.C.); xypan@qztc.edu.cn (X.P.)

**Abstract:** CdS@ZnS core shell nanocomposites were prepared by a one-pot hydrothermal route. The morphology of the composite was tuned by simply changing the Zn$^{2+}$ precursor concentration. To characterize the samples prepared, various techniques were employed, including XRD, FESEM, TEM, XPS and UV-vis DRS. The band gaps of CdS and ZnS were measured to be 2.26 and 3.32 eV, respectively. Compared with pure CdS, the CdS@ZnS samples exhibited a slight blue shift, which indicated an increased band gap of 2.29 eV. The CdS@ZnS core shell composites exhibited efficient photocatalytic performance for H$_2$ generation under simulated sunlight illumination in contrast to pure CdS and ZnS. Additionally, an optimized H$_2$ generation rate (14.44 mmol·h$^{-1}$·g$^{-1}$$_{cat}$) was acquired at CdS@ZnS-2, which was approximately 4.6 times greater than that of pure CdS (3.12 mmol·h$^{-1}$·g$^{-1}$$_{cat}$). Moreover, CdS@ZnS heterojunction also showed good photocatalytic stability. The process of charge separation over the photocatalysts was investigated using photoelectrochemical analysis. The findings indicate that the CdS@ZnS nanocomposite has efficient charge separation efficiency. The higher H$_2$ generation activity and stability for CdS@ZnS photocatalysts can be attributed to the intimate interface in the CdS@ZnS core–shell structure, which promoted the light absorption intensity and photoinduced charge separation efficiency. It is expected that this study will offer valuable insights into the development of efficient core shell composite photocatalysts.

**Keywords:** photocatalyst; chemical state; core shell; H$_2$ production

## 1. Introduction

Hydrogen is considered an important clean and renewable energy carrier [1]. Currently, most existing hydrogen production technologies rely on fossil energy, which is harmful to the environment. Therefore, tremendous effort has been devoted to developing green techniques for H$_2$ production. Among the green techniques available to generate H$_2$, photocatalytic hydrogen generation from water with the merits of environmental friendliness has gained increasing attention for its potential applications in converting solar energy to chemical energy [2]. The key task in photocatalytic hydrogen production is to design photocatalytic materials that exhibit both high efficiency and stability.

Researchers are greatly interested in CdS photocatalysts for H$_2$ generation due to their narrower bandgap (approximately 2.4 eV), which enables visible light response and sufficiently negative potential of the conduction band edge for proton reduction [3]. However, CdS photocatalysts often exhibit unsatisfactory photocatalytic performance due to faster charge recombination and photocorrosion [4]. The efficiency, stability and long-term practicality of hydrogen generation are restricted by these disadvantages.

Several approaches, such as morphology control [5], crystal phase control [6], heterojunction formation [7,8], doping with metals [9] and nonmetals [10] have been employed to overcome these limitations. Nowadays, methods to improve the photocatalytic performance of CdS mainly focus on structural engineering or surface modification. Different nanostructures, including nanowires, nanorods and nanospheres, have been prepared for CdS. However, the achievements in improving photocatalytic activity and stability are not satisfactory. Noble metal ion doping (such as gold, palladium, silver, nickel, etc.) has been proven to be able to improve the activity of photocatalysts. However, it cannot be applied to industrial production because of its high cost. More importantly, CdS photocatalysts are usually unstable in the photocatalytic reaction process. Once exposed to the electrolyte, the Cd-S bond is easily broken by photogenerated hole oxidation ($CdS + 2h \rightarrow Cd^{2+} + S$), which leads to poor photostability [11].

Constructing a core–shell structure is one of the effective methods to inhibit photocorrosion [12]. Generally, core materials should be made up of photo-corrosive materials, and shell structures should contain stable materials. Zinc sulfide (ZnS) is widely recognized as a semiconductor with low toxicity, exhibiting remarkable photoluminescence and exceptional stability. It possesses a broad energy band gap of 3.6 eV. Therefore, ZnS can be used to create protective shells around the CdS core to prevent the photo-corrosion of chalcogenides [13]. In addition, ZnS crystal contains many inherent imperfections that can function as hole acceptors for CdS to effectively impede photo-corrosion [14]. The core shell material can rapidly transfer carriers through a close interface, which effectively reduces photo-corrosion. As a result, the surface recombination sites of the CdS core may be greatly minimized. Thereby, the photocatalyst efficiency and photostability are improved [15].

Recently, the CdS/ZnS core shell nanostructures have been reported by several researchers. Wang et al. described a three-step chemical synthesis to prepare ZnS/CdS core -shell nanotubes. Compared to both pure ZnS nanotubes and CdS nanotubes, ZnS/CdS core shell nanotubes exhibit significantly higher activity in photocatalytic hydrogen production [16]. Su and Shim et al. developed a simple approach involving two steps for the preparation of CdS/ZnS core shell microparticles, which demonstrated improved photocatalytic properties compared to individual CdS or ZnS particles [17,18]. However, CdS/ZnS synthesized through a multistep method usually presents a noticeable boundary between the two photocatalysts, leading to limited interaction, which is unfavorable for the separation of photogenerated charges [19].

In particular, several single-step solvothermal preparation methods have been reported. According to Zhang and colleagues, a significant improvement in photocatalytic activity was achieved by synthesizing CdS/ZnS core shell nanocrystals conjugated with pectin using a microwave-assisted method [13]. In addition, Zhang et al. prepared polyacrylamide (PAM) colloid particles containing CdS/ZnS core shell nanoparticles [20]. Furthermore, CdS/ZnS core shell nanoparticles were deposited on montmorillonite (MMT) by Kamilla et al. [21]. However, those organic surfactants and modifiers may inevitably contaminate the heterostructures. Furthermore, the fabrication methods are not very environmentally friendly [22].

Based on the above considerations, we have successfully produced a CdS@ZnS core–shell structure with different ZnS contents using a surfactant-free one-pot hydrothermal method, which exhibits higher photocatalytic hydrogen evolution performance and stability under simulated sunlight. An intimate interface between the ZnS shell and CdS core was formed. As a result, photoexcited charge carriers can be separated efficiently. Consequently, the optimized CdS@ZnS presented a significantly improved rate and durability in the photocatalytic production of $H_2$. It is hoped that this simple surfactant-free solvothermal method to prepare core–shell heterostructures may inspire the synthesis of other semiconductor heterostructures.

## 2. Results and Discussion

*Characterization of CdS@ZnS Samples*

Material structures and morphologies have a significant impact on their performance. The crystal phases of the ZnS, CdS and CdS@ZnS samples were characterized using XRD. As depicted in Figure 1, for the blank ZnS, the characteristic peaks of ZnS at 28.6°, 47.6° and 56.5° were attributed to the cubic ZnS crystal planes (111), (220) and (311) (PDF#65-0309), respectively [23]. For the blank CdS, the CdS characteristic peaks located at 24.8°, 26.5°, 28.2° and 43.7° were attributed to the crystal planes (100), (002), (101) and (110) of hexagonal CdS (PDF#89-2944 CdS), respectively [24]. For CdS@ZnS samples, both the characteristic peaks of CdS and ZnS could be observed [18]. In addition, the XRD peaks of CdS shifted toward a higher angle with the decoration of ZnS nanoparticles. According to previous reports, as defects formed, the XRD peaks will shift toward higher angles [25]. Defects tend to be formed at the interface region of the nanocomposite due to the lattice mismatch between the two components. Therefore, the shift in CdS XRD peaks may be ascribed to the formation of interfacial defects after ZnS decoration.

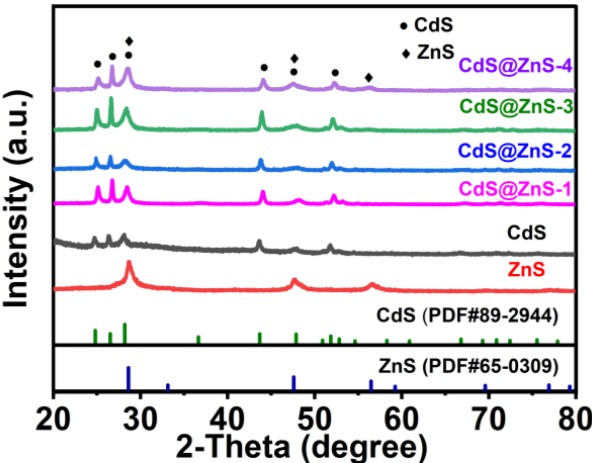

**Figure 1.** XRD patterns of the ZnS, CdS and CdS@ZnS samples.

Because the crystal structures of cadmium sulfide and zinc sulfide are different, they will not form a solid solution during the hydrothermal process. Because the lattice spacing between the ZnS (111) crystal plane and CdS (101) crystal plane is very close, it is possible for ZnS nanocrystals to grow on the surface of CdS crystal. Eventually, CdS@ZnS nanostructures would be formed.

The microscopic morphologies of the samples were analyzed using FESEM and TEM. CdS nanoparticles were self-assembled into spherical structures with a smooth surface (Figure 2a). For the preparation of CdS@ZnS nanocomposite, $S^{2-}$ was produced by the decomposition of $NH_2CSNH_2$ precursor at a high temperature. The $K_{sp}^{\theta}$ constant of CdS ($8.0 \times 10^{-27}$) is much lower than that of ZnS ($2.5 \times 10^{-22}$) [26]. Therefore, Cd $(CH_3COO)_2 \cdot 2H_2O$ would react with dissociated $S^{2-}$ to form CdS first in the solution. Then, the excess $S^{2-}$ would further react with $Zn(Ac)_2 \cdot 2H_2O$. The interplanar mismatch between CdS and ZnS is very small. Therefore, ZnS is deposited onto the surface of CdS via heterogeneous nucleation and growth. ZnS possesses a different crystal structure from that of CdS. $Zn_xCd_{1-x}S$ solid solution does not form during synthesis [27]. Therefore, the CdS NPs acted as the centers for the lateral deposition of tiny ZnS NPs. After the attachment of tiny ZnS NPs, the surface morphology of CdS@ZnS NPs had flower-like appearances. Interestingly, the surface became rougher with the increase in the concentration of the $Zn^{2+}$ ion precursor (as shown in Figure 2b–e). The coarse surfaces were formed of self-aggregated tiny ZnS NPs.

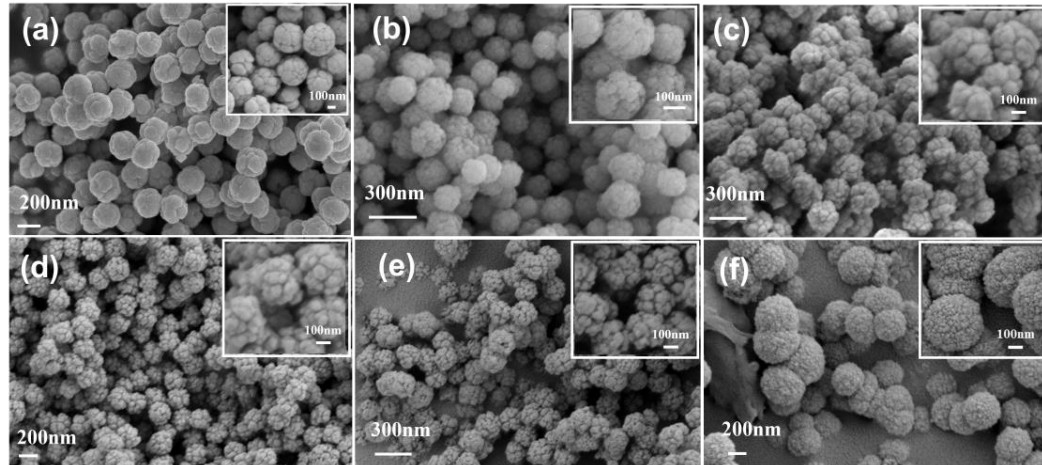

**Figure 2.** SEM images of (**a**) CdS, (**b**) CdS@ZnS-1, (**c**) CdS@ZnS-2, (**d**) CdS@ZnS-3, (**e**) CdS@ZnS-4 and (**f**) ZnS.

TEM, HRTEM and high-angle annular dark-field scanning transmission electron microscopy (HAADF STEM) elemental mapping were performed to further examine the structure and morphology of pure CdS and CdS@ZnS-2. Compared to the spherical CdS NPs, it was evident that CdS@ZnS samples were decorated with a new layer (as shown in Figure 3a,d). A closer examination of the edges of the products in Figure 3b,e implies the formation of tiny ZnS NPs onto CdS cores, causing the surfaces of the CdS@ZnS samples to roughen.

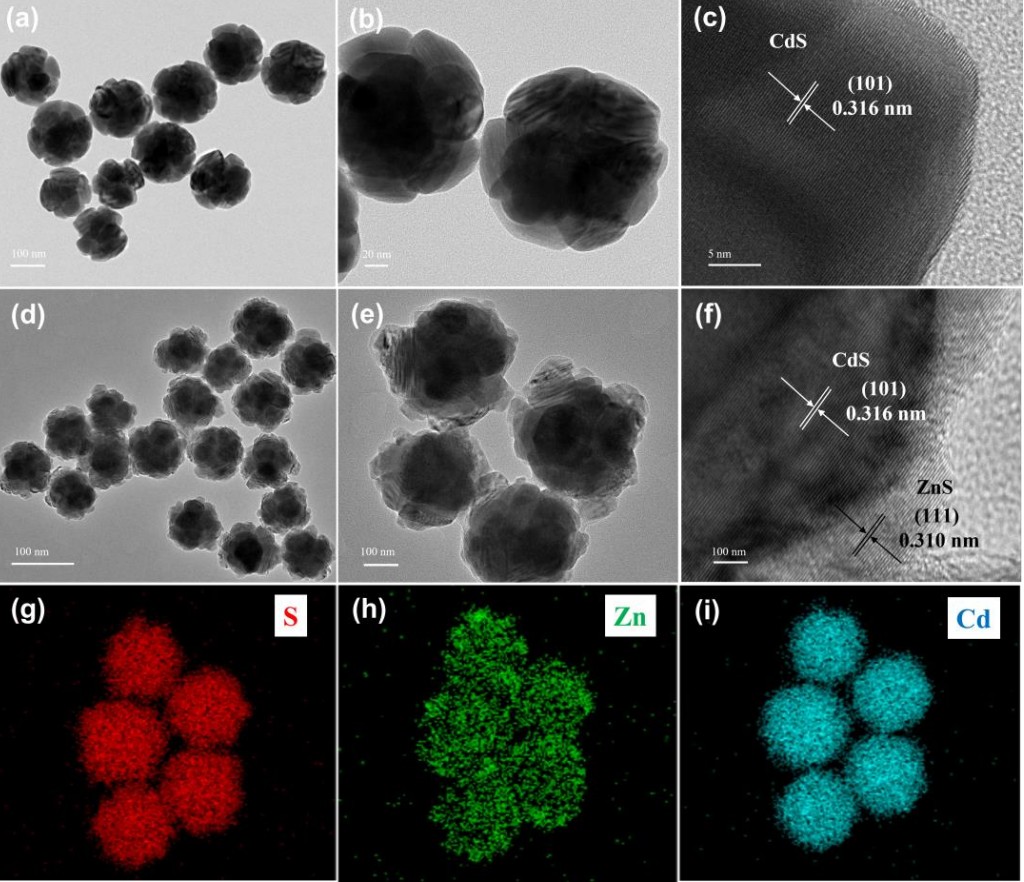

**Figure 3.** (**a**,**b**) TEM images and (**c**) HRTEM image of pure CdS NPs; (**d**,**e**) TEM images and (**f**) HRTEM image of CdS@ZnS-2; elemental maps of (**g**) S, (**h**) Zn and (**i**) Cd.

The corresponding HRTEM images in Figure 3c,f further confirm the above results. The interplanar spacing of the CdS NPs was about 0.316 nm, which corresponded to the wurtzite-structured CdS (101) plane. The HRTEM image of tiny ZnS NPs showed 0.310 nm interlayer lattice spacings, which matched well with the (111) planes of cubic CdS. The small interplanar mismatch between CdS (101) and ZnS (111) is advantageous for maintaining the thermal stability of nanoheterostructures [22]. The core–shell structure of CdS@ZnS was further confirmed by HAADF-STEM elemental mapping, as presented in Figure 3g,h,i. Element S was detected through the whole particle. The Cd element was mainly distributed on the core structure. In contrast, Zn was abundant at the surface and surrounded the dense Cd core, suggesting successful fabrication of the core–shell structure [28]. The results were further verified by the EDS data (Figure S1). This indicates that the main elements on the surface of CdS@ZnS-2 were Cd, Zn and S. The weight percentages of Cd, S and Zn were determined to be 59.5%, 33.6% and 6.9%, respectively, by EDS analysis.

XPS analysis was explored to understand the chemical states of the CdS@ZnS composite. Figure 4 displays the XPS spectra of as-prepared CdS@ZnS-2. Figure 4a shows that C, O, Cd, Zn and S were present in the full XPS spectrum of CdS@ZnS-2 composite. CdS@ZnS-2 XPS spectrum of Zn 2p is presented in Figure 4b. The binding energy peaks at 1022.7 eV belong to Zn $2p_{3/2}$, while another peak located at 1045.71 eV corresponds to Zn 2p1/2 for the divalent Zn in ZnS [29].

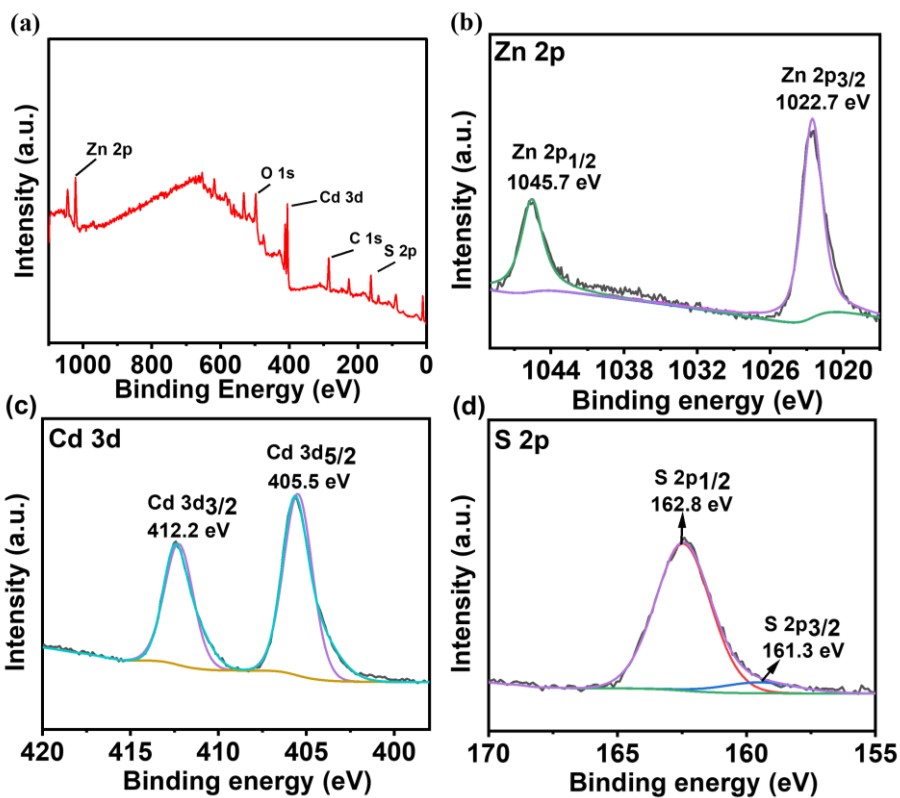

**Figure 4.** XPS spectra of (**a**) survey, (**b**) Zn 2p, (**c**) Cd 3d and (**d**) S 2p for CdS@ZnS-2.

The Cd 3d and S 2p spectra are displayed in Figure 4c,d, respectively. The XPS peak of Cd 3d could be split into two peaks with values of 405.7 eV for Cd $3d_{5/2}$ peak and 412.2 eV for Cd $3d_{3/2}$ peak, respectively (Figure 4c). The doublet characteristic peaks suggest the presence of $Cd^{2+}$ in CdS [30]. As depicted in Figure 4d, the peaks at 162.8 and 161.3 eV were attributed to the spin-orbit states of S $2p_{1/2}$ and S $2p_{3/2}$, respectively [15]. The atomic ratios of S, Cd and Zn were determined to be 41.1%, 38.1% and 20.8%, respectively, by XPS analysis. These results further confirm that the Zn 2p and Cd 3d binding energies matched well with the literature for the $Zn^{2+}$ and $Cd^{2+}$ in pure metal-sulfide [31].

UV-Vis diffuse reflectance spectroscopy (DRS) is utilized to investigate the optical properties of the samples, as shown in Figure 5. The CdS NPs exhibited a sharp incline in absorption at approximately 550 nm. In addition, the band gap of CdS, ZnS and CdS@ZnS samples were obtained via the Kubelka–Munk formula [32].

$$A h\nu = k(h\nu - E_g)^{n/2}$$

where $\alpha$, h, $\nu$, k and $E_g$ are absorption coefficient, Planck constant, photon's frequency, a constant and band gap energy, respectively, and n is determined by the electron transition of the semiconductor. For the direct and indirect transition band gap semiconductors, the values of n were 1 and 4, respectively. However, CdS is a common indirect band gap semiconductor material, so n was 4 in this experiment. The band gap energy ($E_g$) usually depends on diffuse reflectance spectra, which can be calculated from the UV-DRS spectra via plotting $(\alpha h\nu)^2$ against h$\nu$. Therefore, the band gap energy of CdS was 2.26 eV, as estimated by the Tauc plot (Figure 5b) [8]. The pure ZnS sample with a band gap of 3.32 eV exhibited electronic absorption in the UV light region. It was noticeable that the CdS@ZnS samples showed a slight blue shift in comparison with the CdS NPs, indicating an increased band gap of 2.29 eV. Furthermore, the CdS@ZnS samples also exhibited enhanced light absorption intensity in the UV-vis region after the deposition of tiny ZnS NPs, which implies that the sample could have good photoactivity [33].

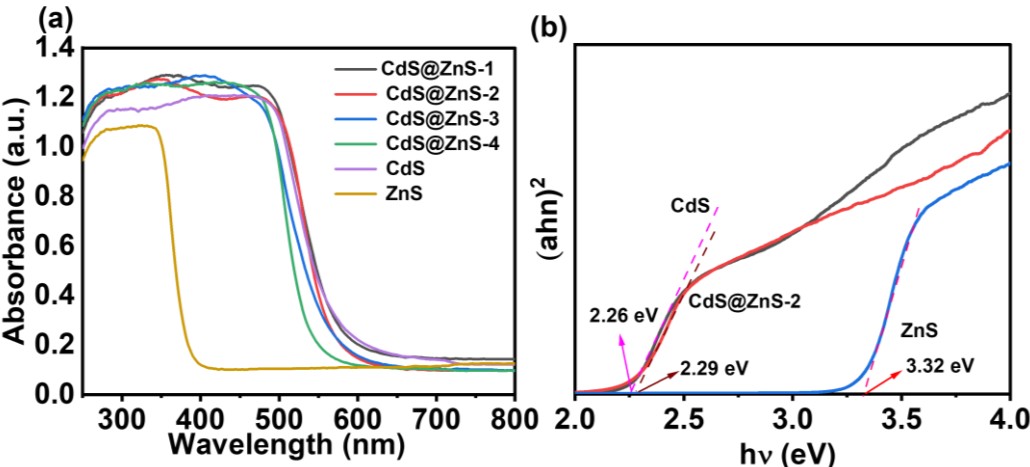

**Figure 5.** (**a**) UV-visible diffuse reflection absorption spectrum of the prepared sample; (**b**) Corresponding Tauc diagrams.

Photocatalytic $H_2$ generation was utilized to measure the photocatalytic activities of CdS, ZnS and CdS@ZnS samples. Under simulated sunlight irradiation, CdS and ZnS showed moderate photoactivity for $H_2$ evolution. According to Figure 6a, all CdS@ZnS photocatalysts exhibited higher $H_2$ generation activity in comparison with pure CdS and ZnS. The result showed that the CdS@ZnS heterostructure effectively improved the photoactivity of the sample. For CdS@ZnS photocatalysts, the rate of $H_2$ evolution was measured to be 11.68, 14.44, 7.18 and 4.96 mmol·$h^{-1}$·$g^{-1}_{cat}$ for CdS@ZnS-1, CdS@ZnS-2, CdS@ZnS-3 and CdS@ZnS-4, respectively [34]. The higher $H_2$ generation activity for CdS@ZnS photocatalysts may ascribe to the stronger UV-vis light absorption intensity after the deposition of tiny ZnS NPs according to the DRS analysis [35]. The optimized $H_2$ generation rate (14.44 mmol·$h^{-1}$·$g^{-1}_{cat}$) was acquired at a Cd and Zn molar ratio of 7:3 in CdS@ZnS-2, which was approximately 4.6 times enhanced than that of pure CdS (3.12 mmol·$h^{-1}$·$g^{-1}_{cat}$). The higher $H_2$ generation activity and stability for CdS@ZnS-2 photocatalysts can be attributed to the intimate interaction in the CdS@ZnS core–shell structure, which promoted the photoinduced charge separation efficiency. The interfacial charge transport properties were also confirmed by an electrochemical impedance spectroscopy (EIS) study.

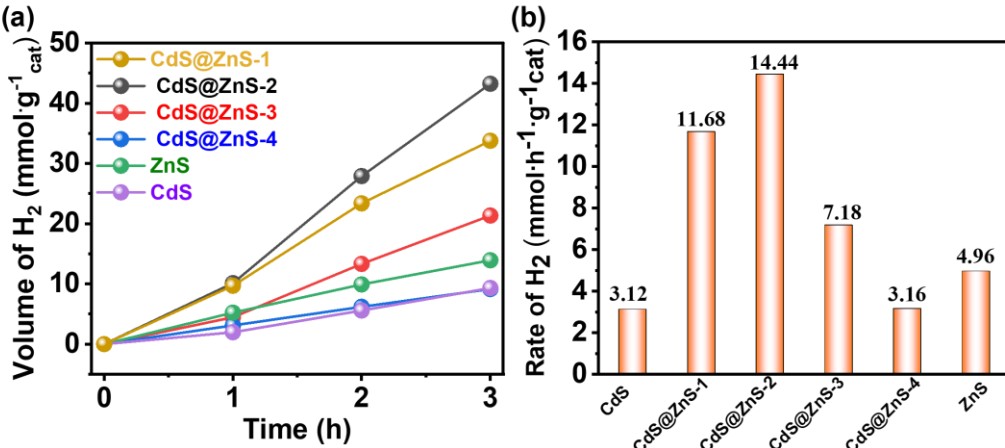

**Figure 6.** (**a**) Photocatalytic hydrogen production activity and (**b**) hydrogen production rate diagram of the prepared sample.

As CdS is known to corrode easily, the cyclic stability test of photocatalytic hydrogen production activity for CdS@ZnS-2 sample was performed for 3 cycles under simulated sunlight (Figure 7). After three cycles of photocatalytic reaction, the activity of the sample was almost unchanged, which suggests excellent stability of the photocatalyst.

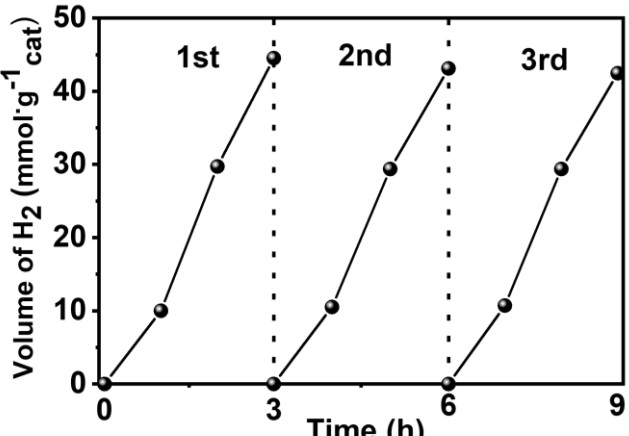

**Figure 7.** Cyclic stability diagram of photocatalytic hydrogen production activity for CdS@ZnS-2 sample under simulated sunlight.

Moreover, the XRD and SEM analysis were utilized to measure the crystal phase and microscopic morphology of CdS@ZnS-2 after the recycling test. The XRD diffraction peaks of the CdS@ZnS-2 before and after the recycle test are compared in Figure 8a. The XRD diffraction peaks were almost unchanged after the photocatalytic reaction. In addition, there was no significant change in the morphology of CdS@ZnS-2 after the reaction, which was similar to the original CdS@ZnS-2 photocatalyst.

Furthermore, the surface analysis of CdS@ZnS-2 after the recycling test was measured by XPS analysis in Figure S2. The XPS spectra of Zn 2p, Cd 3d and S 2p showed no significant change in peak position and intensity, suggesting good stability of the as-prepared photocatalyst.

The highly efficient photoactivity of CdS@ZnS-2 was primarily due to the efficient charge separation, which was further investigated by electrochemical impedance spectroscopy (EIS) study (Figure 9). The Nyquist plot of CdS@ZnS-2 showed the smallest arc radius compared to those of the other samples, demonstrating the lowest interfacial charge-transfer resistance [36]. This result was consistent with its highest photoactivity. Further

increasing the ZnS content would increase the charge transfer resistance. In addition, the wide bandgap ZnS with increased content would limit the light absorption.

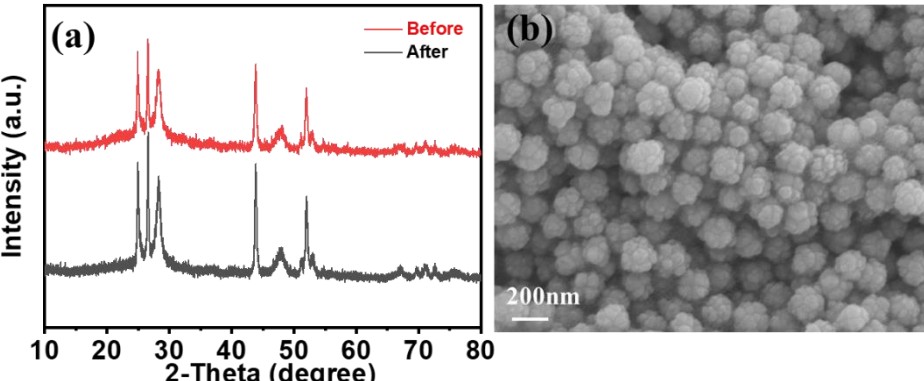

**Figure 8.** (**a**) XRD of CdS@ZnS-2 before and after the recycling test; (**b**) SEM of CdS@ZnS-2 after recycling test.

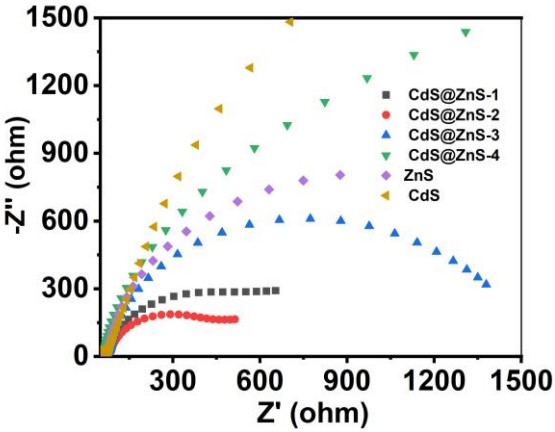

**Figure 9.** EIS Nyquist diagram of CdS and CdS@ZnS catalysts.

The photoluminescence (PL) spectra can reflect the recombination of carriers generated by light irradiation. Figure S3 shows the photoluminescence spectra of CdS@ZnS-2 core/shell structures and pure CdS. The 540 nm peak in the PL spectra was ascribed to the band edge emission of CdS [17]. Compared to pure CdS, the intensity of this peak was obviously lower than that of CdS@ZnS-2 due to the efficient suppression of excited carriers through photoluminescence emission [37].

The schematic illustration of the photo-induced charge separation process for the production of hydrogen in the CdS@ZnS photocatalysts was displayed in Figure 10. The CdS@ZnS is similar to a type-II heterojunction, facilitating the transfer of photo-induced electrons from ZnS to CdS [38]. From the band alignment, it was observed that CdS had a lower C.B. edge potential than ZnS. The photo-induced electrons in the ZnS shell can be readily transferred to the CdS core. The photogenerated electron could be captured by $H^+$ ions to produce $H_2$. Compared to CdS, ZnS has a deeper V.B. maximum.

The transfer of photoexcited holes from CdS to ZnS seems unfavorable. However, previous reports have shown that the crystal lattices of ZnS often have zinc vacancies ($V_{Zn}$) and interstitial sulfur vacancies ($I_S$). The zinc vacancies ($V_{Zn}$) and interstitial sulfur vacancies ($V_S$) can function as acceptor states to capture the photoexcited holes from CdS [28,39–41]. Subsequently, the photoexcited holes are captured by lactate ions [42]. According to a previous report, the possible products of lactic acid oxidation are $CO_2$, acetaldehyde and pyruvic acid [42].

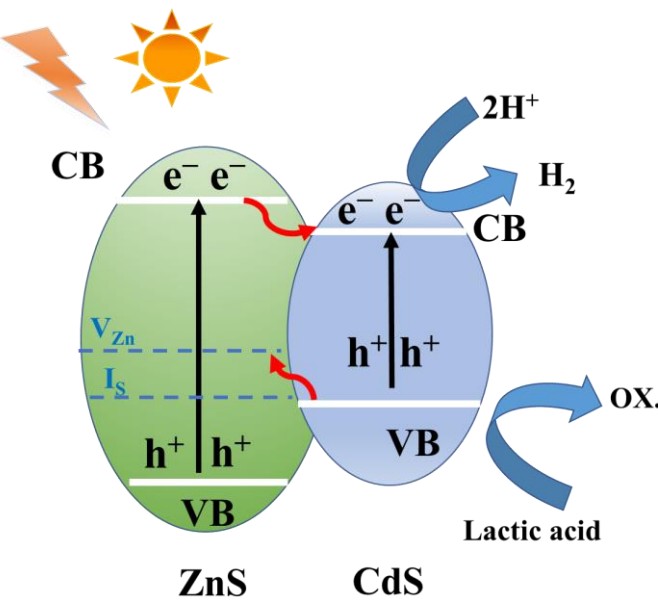

**Figure 10.** CdS@ZnS band structure mechanism of composite materials.

Thus, this favorable feature of the band structure is advantageous for the spatial separation of photoexcited carriers and is attributed to the improved photocatalytic hydrogen production performance from the CdS@ZnS heterostructure [43].

## 3. Materials and Methods

### 3.1. Materials

Cadmium acetate dihydrate (Cd $(CH_3COO)_2 \cdot 2H_2O$), zinc acetate dihydrate ($Zn(Ac)_2 \cdot 2H_2O$) and thiourea ($NH_2CSNH_2$) were obtained from the Sinopharm Chemical Regent Co., Ltd. (Shanghai, China) without additional purification. All reagents were analytically pure. Deionized water was supplied by local sources.

### 3.2. Synthesis

#### 3.2.1. Synthesis of CdS

A simple solvothermal method was used to synthesize CdS. For a standard process, Cd $(CH_3COO)_2 \cdot 2H_2O$ (0.2665 g) and $NH_2CSNH_2$ (0.9135 g) were completely dissolved in 80 mL deionized water. Following 30 min of vigorous stirring, the solution was moved to a Teflon-lined autoclave with a capacity of 100 mL. Then, the reactor was kept at 160 °C for 12 h under autogenous pressure. The yellow precursors were collected through centrifugation after they cooled down to room temperature. Then, the as-prepared sample was washed multiple times with deionized water and anhydrous ethanol. Finally, the samples were dried at 60 °C for 12 h.

#### 3.2.2. Synthesis of ZnS

The ZnS nanomaterials were prepared as follows. Typically, zinc acetate and thiourea are mixed at a molar ratio of 1:1. Afterwards, the solution was dissolved in water. The solution was intensely stirred for 30 min. Next, the mixture was moved into a 50 mL Teflon reaction kettle and hydrothermally treated at 160 °C for 12 h. The resultant suspension was washed three times with deionized water and subsequently dried in an oven at a temperature of 60 °C.

#### 3.2.3. Synthesis of CdS@ZnS

CdS@ZnS nanocomposites with different molar rations of Cd to Zn were synthesized via a one-step hydrothermal route. The synthesis of CdS@ZnS-1 with molar ratio of Cd to Zn of 9:1 was shown as follows. Briefly, Cd $(CH_3COO)_2 \cdot 2H_2O$ (0.2665 g), $NH_2CSNH_2$

(0.9135 g) and $Zn(Ac)_2 \cdot 2H_2O$ (0.0549 g) (Cd: Zn = 9:1) were dissolved in 80 mL of deionized water. Following 30 min of vigorous stirring, the solution was moved to a Teflon-lined autoclave with a capacity of 100 mL. Then, the reactor was kept at 160 °C for 12 h under autogenous pressure. The yellow precursors were collected through centrifugation after they cooled down to room temperature. Then, the as-prepared sample was washed multiple times with deionized water and anhydrous ethanol. Finally, the samples were dried at 60 °C for 12 h. The obtained products were denoted as CdS@ZnS-1 with a molar ratio of Cd to Zn of 9:1. CdS@ZnS with molar ratios of Cd to Zn of 8:2, 7:3 and 5:5 were prepared by the same method, recorded as CdS@ZnS-2, CdS@ZnS-3 and CdS@ZnS-4.

### 3.3. Characterization

Rugaku Miniflex II X-ray diffraction was utilized to investigate the crystal structures of the samples. X-ray diffraction was performed using a Cu $K_\alpha$ target with a 40 kV voltage, 40 mA current, step width of $0.02° \cdot 2\theta \cdot s^{-1}$ and a scan range of 20°–80°. The morphology and microscopic structure were determined using a field-emission scanning electron microscope (FESEM, JSM-6700F, JEOL Co., Ltd., Tokyo, Japan) and transmission electron microscope (JEM-2010, FEI, Tecnai G2 F20 FEG TEM, FEI Co., Hillsboro,USA) operated at $200 \times 10^3$ V. X-ray photoelectron spectroscopy (XPS) measurements (Thermo Scientific ESCA Lab250, Thermo Fisher Scientific Inc., Madison, WI, USA) were performed to obtain the chemical composition and the valence state with Al $K_\alpha$ as the X-ray source (1486.6 eV). The peaks were adjusted according to the standard C 1s (284.6 eV) for calibration. UV-visible ultraviolet/visible diffuse reflectance spectra (PerkinElmer Lambda 950 UV/VIS/NIR) were used to characterize the optical properties. The photoluminescence (PL) of the samples was carried out on an F-7000 Fluorescence Spectrophotometer.

### 3.4. Photoelectrochemical Measurements

The photoelectric–chemical analysis was carried out using a conventional three-electrode cell (electrochemical workstation: CHI660C Instruments, CH instruments, Inc., Austin, TX, USA). A 300 W Xe lamp was utilized as the light source. An Ag/AgCl electrode and Pt electrode were utilized as the reference electrode and counter electrode, respectively. The work electrodes were prepared in accordance with our previous publication [44]. First, the FTO glasses were sonicated in ethanol and deionized water for 30 min. Next, the FTO glasses were dried at 60 °C for 3 h. Second, 1 mL Nafion solution and 1 mL ethanol solution were mixed with 8 mL $H_2O$ to obtain solution A. A 0.01 g sample was dispersed in 500 mL solution A. After sonication for 30 min, the mixed solution was spread to the conductive surface of the FTO glasses to form a film with 0.25 cm$^2$ active area. Then, the FTO glasses were dried at 60 °C for 12 h. FTO glasses were used as the working electrode. A 0.2 mol/L $Na_2SO_4$ solution was used as the electrolyte. Mott–Schottky investigations were carried out in a CHI660C workstation with homemade three-electrode quartz cells. The electrochemical impedance spectra (EIS) were conducted using a 0.01 mol/L $K_3[Fe(CN)_6]/K_4[Fe(CN)_6]$ (1:1) mixed solution. Among them, the electrode voltage is −0.5 V, the frequency range was from 0.1 kHz to 100 kHz, and the modulation amplitude of the open circuit voltage was 5 mV.

### 3.5. Photocatalytic Activities

Under irradiation of a 300 W Xe lamp, photocatalytic hydrogen generation by splitting water was carried out in a 500 mL glass-enclosed gas circulation system. Typically, a 5 mg photocatalyst was spread uniformly in 25 mL 20% sacrificial agent solution under magnetic stirring. To eliminate any dissolved oxygen, the system was degassed for 30 minutes before the irradiation process. Next, a 300 W Xe lamp was turned on. The 300 W Xe lamp was placed 20 cm away from the reactor.

The cooling water circulation system was employed to keep the reaction temperature at about 5 °C throughout the experiment, and the reaction solution was stirred by a magnetic

stirrer to ensure that the reaction catalyst was fully dispersed in the mixed solution all the time. The total reaction time was three hours. Hydrogen was detected every hour.

Gas chromatography (GC9790IIFuli, Fuli Analytical Instruments Co., Taizhou, China) with a thermal conductivity detector (TCD) was used to analyze the products.

## 4. Conclusions

In conclusion, we successfully employed a one-step hydrothermal technique for the in situ growth of ZnS shells on CdS cores. By modifying the precursor concentrations in aqueous solution, CdS@ZnS core–shell structures with different compositions were synthesized. FESEM and TEM analysis confirmed the CdS@ZnS core–shell structure. The CdS NPs acted as the centers for the deposition of tiny ZnS NPs. The surface morphology of CdS@ZnS NPs had flower-like appearances. The UV-Vis diffuse reflectance spectroscopy (DRS) indicated that the CdS@ZnS samples showed a slight blue shift with enhanced light absorption intensity in the UV-vis region, which was one of the reasons for the improvement of photocatalytic activity. Under simulated sunlight illumination, the as-synthesized CdS@ZnS core–shell heterostructures exhibited improved photocatalytic performance and stability for $H_2$ production. Compared to pure CdS and ZnS, the CdS@ZnS-2 catalyst showed the highest $H_2$ generation rate of 14.44 mmol·$h^{-1}$·$g^{-1}_{cat}$ and a photocatalytic stability over 9 h. The photoelectric measurements further confirmed the efficient separation and transfer of photoinduced charge carriers in the CdS@ZnS core–shell heterostructures. Relying on the low lattice mismatch, the immediate interface contact between CdS and ZnS can be attributed to the enhanced photocatalytic activity and stability. It is hoped that this simple surfactant-free solvothermal technique to synthesize core shell heterostructures may inspire the synthesis of other semiconductor heterostructures.

**Supplementary Materials:** The following supporting information can be downloaded at: https://www.mdpi.com/article/10.3390/catal13071123/s1, Figure S1: selected area EDS spectra of the CdS@ZnS-2 sample; Figure S2: XPS spectra of (a) survey, (b) Zn 2p, (c) Cd 3d, and (d) S 2p for CdS@ZnS-2 after recycling test; Figure S3: PL spectra of CdS and CdS@ZnS-2 catalysts.

**Author Contributions:** Conceptualization, M.H. and X.P.; methodology, M.H. and M.Y.; validation, M.H., S.C. and X.P.; formal analysis, M.H. and M.Y.; investigation, M.H.; data curation, M.Y., R.S., X.Z., S.C. and K.L.; writing—original draft preparation, M.H. and M.Y.; writing—review and editing, M.H. and X.P.; visualization, M.H.; supervision, X.P.; project administration, X.P.; funding acquisition, X.P. All authors have read and agreed to the published version of the manuscript.

**Funding:** This research was funded by the Natural Science Foundation of Fujian Province (2021J05181, 2021J01966, 2021J01470), the Program for New Century Excellent Talents in Fujian Province University, the Award Program for Tongjiang Scholar Professorship, the Innovation and Entrepreneurship Projects for High-level Talents of Quanzhou (2017Z028), and the Quanzhou Science and Technology Project (Grant No. 2021C023R).

**Data Availability Statement:** Data available on request due to restrictions eg privacy or ethical. The data presented in this study are available on request from the corresponding author.

**Conflicts of Interest:** The authors declare no conflict of interest.

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
