# Peer review of "Tailoring Morphology in Hydrothermally Synthesized CdS/ZnS Nanocomposites for Extraordinary Photocatalytic H2 Generation via Type-II Heterojunction"

_catalysts, doi:10.3390/catal13071123_

Round 1

Reviewer 1 Report

This manuscript reports the synthesis of tunable CdS@ZnS composites for H2 production. The productions reported seem promising, and the results were well distributed in the manuscript. Thus, I recommend its publication after minor corrections. Please, consider the following observations:

Introduction

- Include more discussion about how the presence of ZnS could limitate the photocorrosion of CdS.

Results

- More analysis of the possible distortion of the structure or the changes in the crystallite size should be included.

- Page 3, lines 116-117, expand this information.

- It needs to be made obvious the growth of each component in the SEM images; please, include EDS mappings i. Why the Zinc appears in the bulk if the analysis showed that it is outside of the core, include more analysis to prove that information.

- Figure 4, It is very important that the authors re-make the deconvolution analysis. Please, carefully revise the baseline in all the spectra (mainly in the Zn 2p).

- Include more explanation about why the sample -2 was better to evolute H2.

- What are the possible products of the lactic acid oxidation?

- Include the surface analysis of the sample after the tests (not only XRD or SEM).

General

- Include references about earth-abundant S-based materials: https://doi.org/10.1016/j.mssp.2021.106029

https://doi.org/10.1016/j.ijhydene.2021.09.007

Minor revisions required.

Reviewer 2 Report

The authors reported the one pot hydrothermal synthesis of CdS@ZnS nanocomposites for highly efficient photocatalytic hydrogen production. The manuscript was well organized and the results presented well. I my opinion the manuscript can be accepted for publication in the present form. 

Author Response

Thank you very much for your positive comments.

Reviewer 3 Report

The manuscript "The authors design the entitle hydrothermal synthesis CdS/ZnS nanocomposites with tunable morphology for efficient Photocatalytic H2 generation, is interesting work, However, the comments required to the address the below comments.

1. The authors should change the title as follows Unleashing Solar Alchemy: Tailoring Morphology in Hydrothermally Synthesized CdS/ZnS Nanocomposites for Extraordinary Photocatalytic H2 Generation via Type-II Heterojunction

2. The author should include a few effective keywords such as photocatalyst, chemical state, core-shell, photo-induced and photoelectrochemical. Add introduction following articles (https://doi.org/10.1016/j.ijhydene.2023.01.059)

3. Similar font sizes in figures Arial or the time of the new Raman? Especially Figures 1, 2, 5, and 10

4. Figure.10 redrawn the mechanism of CdS/ZnS core-shell type-II Heterojunction for hydrogen generation. Cite the following articles at appropriate places. https://doi.org/10.1016/j.ijhydene.2018.10.054

5. Author considers Fermi energy levels please verify this article following articles and cite the appropriate places

Figure 6a scale required to changes the d-axis amount of H2 (µmole. g-1cat) and Figure 6b (mmol.h-1. g-1cat) not correlated this, the authors carefully check it. ( https://doi.org/10.1016/j.apcatb.2019.04.090) check this article

6. Figure. 7 required to change the y-axis, and verify the following articles.   

The prepared manuscript plagiarism report -45%, The Athour should revise the less than 20% 

Author Response

Thank you for your valuable comments. Please seethe attachment for details.

Round 2

Reviewer 3 Report

The Authors addressed all queries as well, The present form accepted